# Interactions between physicians and the pharmaceutical industry generally and sales representatives specifically and their association with physicians' attitudes and prescribing habits: a systematic review

Freek Fickweiler, Ward Fickweiler, Ewout Urbach

Crowd for Cure, Groningen, Groningen, The Netherlands

**Correspondence to**
Freek Fickweiler;
freek@crowdforcure.com

## ABSTRACT

**Objectives** The objective of this review is to explore interactions between physicians and the pharmaceutical industry including sales representatives and their impact on physicians' attitude and prescribing habits.

**Data sources** PubMed, Embase, Cochrane Library and Google scholar electronic databases were searched from 1992 to August 2016 using free-text words and medical subject headings relevant to the topic.

**Study selection** Studies included cross-sectional studies, cohort studies, randomised trials and survey designs. Studies with narrative reviews, case reports, opinion polls and letters to the editor were excluded from data synthesis.

**Data extraction** Two reviewers independently extracted the data. Data on study design, study year, country, participant characteristics, setting and number of participants were collected.

**Data synthesis** Pharmaceutical industry and pharmaceutical sales representative (PSR) interactions influence physicians' attitudes and their prescribing behaviour and increase the number of formulary addition requests for the company's drug.

**Conclusion** Physician–pharmaceutical industry and its sales representative's interactions and acceptance of gifts from the company's PSRs have been found to affect physicians' prescribing behaviour and are likely to contribute to irrational prescribing of the company's drug. Therefore, intervention in the form of policy implementation and education about the implications of these interactions is needed.

## INTRODUCTION

The relationship between physicians and the pharmaceutical industry has evoked heated debate for many decades.[1] In 2012, the pharmaceutical industry spent $89.5 billion on physician–pharmaceutical sales representative (PSR) interactions that accounted for 60% of the global sales and marketing spending.[2–8] Previous reports have demonstrated that

PSRs may influence prescribing behaviour.[9–16] However, the evidence determining whether pharmaceutical industry and PSRs interactions influence physicians is divided and contradictory. Studies have indicated that physicians may be unable to distinguish between promotional information and scientific evidence.[17 18] Physicians, however, believe their colleagues are more susceptible to pharmaceutical industry marketing strategies than themselves.[19–22] The majority of the physicians do not believe that they are affected by pharmaceutical industry and PSR interactions. Most medical and governmental institutions have installed guidelines and self-regulatory and legislative checks to regulate the relationship between physicians and the pharmaceutical industry and its representatives.[5 15 16 23–26] However, while administrative proposals for deregulatory reforms

**Figure 1** PRISMA flow diagram showing search strategy and included studies. PRISMA, Preferred Reporting Items for Systematic Reviews and Meta-Analyses.

that would remove some governmental authority over the industry are increasing, scientific evidence rigorously examining the extent of interactions between physicians and pharmaceutical industry and it PSRs is needed. This review evaluates critically and systemically the evidence on the impact of pharmaceutical industry and PSR interactions on physicians.

## METHODOLOGY
### Protocol
We followed a detailed methodology that we described in our review protocol, which is available on request to the corresponding author. Two independent reviewers assessed selected articles as per inclusion/exclusion criteria, shortlisted them for writing the review and cross-checked their decisions about inclusion/exclusion with each other. The review followed the Preferred Reporting Items for Systematic Reviews and Meta-Analyses guidelines (supplementary appendix 1).

### Eligibility criteria
The eligibility criteria were:
- types of studies: observational study design, such as cross-sectional studies and cohort studies, but also (non-)randomised trials and survey designs comparing at least one facet that are mentioned below on the impact on behaviour and attitude;

- types of participants: physicians, pharmaceutical representatives and physicians in training/residents;
- types of exposure: any type of interaction between physicians and the pharmaceutical industry where there is direct interaction with the physician, such as meeting with drug representatives, participating in pharmaceutical-sponsored Continuing Medical Education (CME) events, receiving travel funding, free drug samples, industry-provided meals, gifts and presentations of industry-related information;
- types of outcome: knowledge, beliefs and/or attitudes of physicians regarding physician–industry interactions and (prescribing) behaviour of physicians;
- type of control: no interaction.
- Exclusion criteria were: qualitative, ecological, econometric studies, editorials, letters to the editor, studies on other health professionals (eg, nurses and medical students), small samples sizes, studies assessing indirect interactions and research funding.

We did not exclude studies based on risk of bias. We took risk of bias into account when grading the quality of evidence using GRADE approach.

### Search strategy
The search strategy included PubMed, Embase, Cochrane Library and Google Scholar electronic databases (January 1992 to August 2016). Databases were not searched

before 1992, as these studies were already investigated in an earlier review.[27] The search combined terms for physicians and pharmaceutical and included both free-text words and medical subject heading relevant to the topic. We did not use a search filter. The online supplementary information file provides the full details for one database. Additional search strategies included a search of the grey literature (theses and dissertations). Also, we reviewed the references lists of included and relevant papers.[27–29]

### Assessment of risk of bias in included studies

Two reviewers assessed in duplicate and independently the risk of bias in each eligible study. Disagreements were resolved by discussion or adjudication by a third reviewer. We used the recommendations outlined in the Cochrane Handbook to assess the risk of bias in randomised studies. We graded each potential source of bias and rated the studies as high, low or unclear risk of bias.

### Data analysis and synthesis

The information extracted from the selected studies included type of study, study design, type of pharmaceutical industry and PSR interaction and type of outcome. We did not conduct a meta-analysis due to the heterogeneity of study design, types of interventions, outcomes assessed and outcome measures used. Instead, we summarised the data narratively. We assessed the quality of evidence by outcome using the GRADE methodology.[30]

## RESULTS

We independently screened the titles and abstracts of the 2170 identified records for potential eligibility. Out of 2170, the full text of 49 eligible citations that matched the inclusion criteria were retrieved and used for qualitative assessment during the writing of the review (figure 1, table 1).

### Characteristics of included studies

The identified studies were published between 1992 and August 2016. Most of the studies included were cross-sectional studies.[1 9–13 19 21 22 31–55] Only two studies were cohort studies,[56 57] three were randomised trials[58–60] and one study was a case–control study.[61]

### Extent of interactions between physicians and the pharmaceutical industry

We found that PSR interactions are a regular feature in the daily lives of physicians across the world.[9–11 13 42 50] Most of the attending physicians and residents have at least one interaction with industry representatives per month.[10 21 22 36 42] The frequency of interactions or gifts offered and accepted varies with private versus public hospital setting and the position of the physicians in the medical hierarchy.[10 13 31 38 42 43 50 58 62] Junior residents received twice as much free drug samples from PSR interactions than senior residents.[10] PSR interactions were significantly higher at the beginning of residency.[13] The majority of programme directors of internal

medicine residencies in the USA allowed PSRs to meet with residents during working hours and permitted PSR sponsorship of conferences.[40] Attending physicians and physician specialists had more PSR interactions and received higher numbers of medical samples and promotional material than residents.[9 42] Participants working in private practice alone or in both sectors were more likely to receive gifts than physicians working in the public sector.[38 42 50] Most common gifts received were medical samples,[9 21 22 31 36 37 42 63] promotional material[9 34 42] invitations for dinners,[9] invitations for CMEs,[22 34] scientific journals[34] and free lunches.[21 37]

### Perspectives of physicians towards PSR interactions

We found that physicians have a positive attitude towards PSRs[1 13 19 20 22 31 32 40 49 58 64] Physicians perceived PSRs as important sources of education and funding,[10 22 32 43 45 46]; while some studies reporting sceptical attitudes about the contribution of PSRs towards teaching and education.[21 36 39 40 49] Conference registration fees, informational luncheons, sponsorship of departmental journal clubs, anatomical models and free drug samples were considered as appropriate gifts.[19 39 51 58] Most of the physicians considered pharmaceutical information provided by PSRs, industry-sponsored conferences and CME events as important instruments for enhancing their scientific knowledge.[22 32 45 46] Compared with senior residents, significantly more junior residents felt that pharmaceutical representatives have a valuable teaching role.[10]

Most studies found that physicians do not believe that PSR interactions impact their prescribing behavoir,[19–13 65 66] while other studies found that there was some extent of influence.[21 22 34 36 37 39 43] In addition, physicians considered their colleagues more susceptible than themselves to PSR marketing strategies.[1 20 21 37 43] There was a strong correlation between the amount of gifts and the belief that PSR interactions did not influence their prescribing behaviour.[10]

### Gifts

We found that better scores on knowledge and attitudes were significantly associated with fewer interactions with representatives and their gifts.[19] Conference registration fees, informational luncheons, sponsorship of departmental journal clubs, anatomical models and free drug samples were considered as appropriate gifts[19 37 47 52] Most of the physicians considered themselves immune to the influence of gifts.[1 10 32 33 35 39 53 58] Most common gifts received were medical samples,[9 21 22 31 36 37 42–44 47] promotional material,[9 34 42 67] invitations for dinners[9] and scientific journals.[34]

### Drug samples

Most of the physicians who accepted drug samples had a positive attitude towards the pharmaceutical representatives.[9 21 22 31 36 37 42 43] Accepting samples lead to higher branded drug prescription rather than generic prescribing.[22 47]

**Table 1** Characteristics of included studies

| Study | Authors | Country | Time frame | Participants, setting | Study design | Interaction | Outcomes |
|---|---|---|---|---|---|---|---|
| 1 | Steinman et al[1] | USA | Spring 1999 | Surveys about attitudes and behaviours towards industry gifts in 105 residents at a university-based internal medicine residency programme | Cross-sectional | PSR interactions, gifts | Most participants (61%) hold positive attitudes towards gifts from industry and PSR interactions and believe they do not influence their own prescribing, but only 16% believed other physicians were similarly unaffected (p<0.0001) |
| 11 | De Ferrari et al[9] | Peru | March 2013 | Questionnaire in 155 faculty and trainee physicians of five different clinical departments working in a public general hospital | Cross-sectional | PSR interactions, medical samples, promotional material, dinners | Positive attitude towards representatives (88.5% of participants). Faculty physicians received a larger amount of medical samples and promotional material and were more prone to believe that gifts and lunches do not influence their prescribing behaviour (42.2% vs 23.6%; p=0.036) |
| 12 | Thomson et al[31] | New Zealand | 1991 | Questionnaire survey of 67 general practitioners | Cross-sectional | Interactions with PSR | Most general practitioners (67%) had a negative attitude towards PSR interaction |
| 13 | Kamal et al[32] | Egypt | July and August 2013 | Interviews with 18 physicians | Cross-sectional | Interaction with PSR | Positive attitude towards PSR interaction |
| 14 | Hodges[10] | Canada | October 1993–February 1994 | Survey in 105 residents of psychiatry | Cross-sectional | Interaction with PSR, drug samples, lunches | Positive attitude towards PSR interaction (56.5% of participants). The more money and promotional items a participant had received, the more likely he or she was to believe that discussions with representatives did not affect prescribing (p<0.05) |
| 15 | Gibbons et al[33] | USA | Not reported | Survey of 392 physicians in two tertiary care medical centres | Cross-sectional | PSR interactions, gifts, samples, travel, lunches | Positive attitude towards PSR interactions, gifts, samples and lunches |
| 16 | Spingarn et al[56] | USA | February 1990 | 75 internal medicine physicians in university medical centre | Retrospective cohort | PSR interaction (teaching) | Attendees inappropriately prescribed PSR speakers drug compared with non-attendees (p=0.029) |
| 17 | Zaki[58] | Saudi Arabia | September–November 2013 | Survey of 250 physicians | Randomised, cross-sectional survey | Conferences, drug samples | Favourable towards promotion |
| 18 | Orlowski et al 1994[57] | USA | 1987–1989 | 10 physicians that were invited for a symposium and tracking the pharmacy inventory usage reports for these drugs before and after the symposia | Cohort | Conference travel | Significant increase in the prescribing pattern of drugs occurred following the symposia (p<0.001) |
| 19 | Scheffer et al[34] | Brazil | 2007–2009 | Survey of 300 physicians prescribing antiretroviral drugs | Cross-sectional | Interaction with representative, drug samples, journals | Frequency of interaction; the majority of (64%) of the physicians had multiple forms of interactions with PSR |

Continued

**Table 1** Continued

| Study | Authors | Country | Time frame | Participants, setting | Study design | Interaction | Outcomes |
|---|---|---|---|---|---|---|---|
| 20 | Brett et al[35] | USA | Not reported | Questionnaire of 93 physicians in a medical school | Cross-sectional | Interaction with PSR | Impact on attitudes; most physicians believed that most of PSR activities do not pose major ethical problems |
| 21 | Gupta et al[36] | India | June–September 2014 | Survey of 81 physicians in single hospital | Cross-sectional | Interaction with representative, drug samples, journals | Impact on prescribing; 61.7% of participants think that PSR has an impact on their prescribing (p=0.0001) |
| 22 | Morgan et al[37] | USA | March 2003 | Survey of 397 obstetrician-gynaecologists | Cross-sectional | Drug samples, promotional material, lunch | Impact on prescribing, positive attitudes; most respondents thought it is proper to accept drug samples (92%), lunch (77%), an anatomical model (75%) or a well-paid consultantship (53%) from PSR |
| 23 | Alosaimi et al[19] | Saudi Arabia | 2012 | Survey of 659 physicians | Cross-sectional | Interaction with PSR | Positive attitude towards PSR interaction |
| 24 | Chren and Landefeld[61] | USA | 1989–1990 | 40 case physicians and 80 control physicians | Case–control | PSR interactions, honoraria, research | Increased prescription of company's drug after PSR interaction, honoraria and research (p<0.001, all) |
| 25 | Randall et al[59] | USA | October 2001 | Intervention group of physicians (n=18) that received education about PSR interaction and control group (n=14) | Controlled trial | Interaction with PSR | The majority of residents found the interactions and gifts useful. Compared with the comparison group, the intervention group significantly decreased the reported number of office supplies and non-educational gifts (p<0.05) |
| 26 | Caudill et al[38] | USA | Not reported | Survey of 446 primary care physicians | Cross-sectional | Interaction with PSR | Significant positive correlation between physician cost of prescribing and perceived credibility, availability, applicability and use of information provided by PSR (p<0.01) |
| 27 | Andaleeb and Tallman[20] | USA | Not reported | 223 physicians in northwestern Pennsylvania | Cross-sectional | Interaction with PSR | Positive attitude towards PSR interaction |
| 28 | Reeder et al[39] | USA | 1991–1992 | 87 residents of emergency medicine | Cross-sectional | Interaction with PSR, gifts | Most participants believed that PSR interaction had no impact on their prescribing |
| 29 | Lichstein et al[40] | USA | January–March 1990 | 272 directors of internal medicine residency programmes | Cross-sectional | Interaction with PSR | Most participants had a positive attitude towards PSR interactions |
| 30 | Brotzman et al[41] | USA | Not reported | Directors of 386 family practice residency programme | Cross-sectional | Interaction with PSR | Majority of programmes do not have guidelines for interaction with PSR |
| 31 | Alssageer and Kowalski[42] | Libya | August–October 2010 | Survey of 608 physicians in public and private practice settings | Cross-sectional | Interaction with PSR, drug samples, printed materials | Positive attitude towards PSR interactions |

Continued

**Table 1** Continued

| Study | Authors | Country | Time frame | Participants, setting | Study design | Interaction | Outcomes |
|---|---|---|---|---|---|---|---|
| 32 | Lieb and Brandtonies, 2010[21] | Germany | 2007 | Survey of 208 physicians (neurology, cardiology and general medicine) | Cross-sectional | Interaction with PSR, drug samples, printed materials, lunches | Frequency and impact on attitudes |
| 33 | Lieb and Scheurich[22] | Germany | 2010–2011 | Survey of 160 physicians in private and public practices | Cross-sectional | Interaction with representative, drug samples, printed materials, CME | High expenditure prescribing; avoidance of industry-sponsored CME is associated with more rational prescribing habits |
| 34 | Lieb and Koch[43] | Germany | May–July 2012 | Survey of 1038 medical students at eight universities | Cross-sectional | Interaction with representative, drug samples, printed materials, lunches | Most participants have contact with the pharmaceutical company; 24.6% of the participants thought gifts would influence their future prescribing behaviour, while 45.1% thought gifts would influence their classmates' future prescribing behaviour (p<0.001) |
| 35 | Brown et al[44] | USA | 2008 and 2013 | 251 directors of family medicine residency programmes | Cross-sectional | Interaction with PSR, gifts, lunches | Negative attitude towards PSR interactions |
| 37 | Rahman et al[45] | Bangladesh | December 2008–January 2009 | Survey of 83 village physicians | Cross-sectional | Interaction with PSR | Impact on their prescribing |
| 38 | Lee and Begley[12] | USA | 2008 | Nationally representative survey of 4720 physicians | Cross-sectional | Gifts | Gifts were associated with lower perceived quality of patient care; an inverse relationship between the frequency of received gifts and the perceived quality of care was observed |
| 39 | Montastruc et al[13] | France | August–October 2011 | Survey among 631 medical residents | Cross-sectional | Interaction with representative | Most participants believed that PSR interaction had no impact on their prescribing; participants who had a more positive opinion were more frequently exposed to PSR (p<0.001) |
| 40 | Klemenc-Ketis and Kersnik[46] | Slovenia | October 2011 | 895 family physicians at the primary level of care | Cross-sectional | Interaction with PSR | Positive effect on knowledge; participants value PSRs' selling and communication skills and trustworthiness highly |
| 41 | Hurley et al[47] | USA | 2010 | 3500 dermatologists | Cross-sectional | Free drug samples | Impact on their prescribing; the provision of samples with a prescription by dermatologists has been increasing over time, and this increase is correlated (r=0.92) with the use of the branded generic drugs promoted by these sample |
| 42 | Makowska[48] | Poland | November–December 2008 | Survey of 382 physicians | Cross-sectional | Gifts | Positive attitude towards PSR interactions |
| 43 | Siddiqui et al[49] | Pakistan | Not reported | Questionnaires of 352 medical students | Cross-sectional | Interaction with representative | Positive attitude towards PSR interaction |

Continued

**Table 1** Continued

| Study | Authors | Country | Time frame | Participants, setting | Study design | Interaction | Outcomes |
|---|---|---|---|---|---|---|---|
| 55 | Workneh et al[50] | Ethiopia | February–March 2015 | Survey of 90 physicians from public and private health facilities | Cross-sectional | Interaction with representative, gifts | Positive attitude towards industry, impact on prescribing behaviour; nearly half of the physicians reported that their prescribing decisions were influenced by PSR |
| 57 | Khan et al[51] | Pakistan | Not reported | Questionnaires in 472 physicians | Cross-sectional | Interaction with representative, gifts | Positive attitude towards PSR interaction |
| 58 | Saito et al[67] | Japan | January–March 2008 | 1417 physicians working in internal medicine, general surgery, orthopaedic surgery, paediatrics, obstetrics-gynaecology, psychiatry and ophthalmology | National survey | Interaction with industry, receipt of gifts, funds, CME, samples | Positive attitude towards PSR and gifts, value information from PSR, interactions higher with physicians who prefer to prescribe brand names |
| 59 | Ziegler[18] | USA | 1993 | 27 physicians working in public and private hospitals | Survey | Accuracy of information provided by PSRs about drugs | Incorrect information often provided by speakers goes unnoticed by physicians |
| 60 | Lurie et al[68] | USA | Not reported | 240 internal medicine faculty physicians in academic medical centres | Survey | Effect of interaction with PSR, free meals, honoraria and research support | Impact on prescribing behaviour and formulary change requests |
| 62 | DeJong et al[52] | USA | August–September 2013 | 279 669 physicians who wrote Medicare prescriptions in any of four drug classes: statins, cardioselective β-blockers, ACE inhibitors and angiotensin-receptor blockers, and selective serotonin and norepinephrine reuptake inhibitors Physicians | Cross-sectional | Industry-sponsored meals | Receipt of industry-sponsored meals was associated with an increased rate of brand name prescription. |
| 63 | Yeh et al[53] | USA | 2011 | All licensed Massachusetts physicians who wrote prescriptions for statins paid for under the Medicare drug benefit in 2011 (n=2444) | Cross-sectional | Effect of industry payment on prescription of branded drugs for cholesterol control | Payment for meals and educational programmes increased prescription of brand name statins. |
| 65 | Bowman and Pearle et al[69] | USA | Not reported | 121 physician attendees | Self-report survey | Effect of CME on prescribing behaviour | Sponsoring company's drugs were favoured during prescription |
| 66 | Fischer et al[65] | USA | November 2006–March 2007 | Multidisciplinary focus groups with 61 physicians | Survey | Effect of industry marketing strategies on prescription and cognitive dissonance of physicians | Most participants reported no PSR impact on their prescribing, value to have ability to evaluate information of PSRs |
| 67 | Chimonas et al[66] | USA | June 2004 | Six focus groups in 32 academic and community physicians | Survey | PSR interactions | Positive attitude towards PSR interaction |
| 72 | Yeh et al[54] | USA | Not reported | 1610 US medical students | Cross-sectional | Interaction with representative, gifts, lunches | Policies separating students from representatives reduced number of interactions |
| 73 | Larkin et al[73] | USA | January 2006–June 2009 | Paediatricians, child and adolescent psychiatrists in five medical centres | Survey | Interaction with PSR | Antidetailing policies reduced the prescription of off-label antidepressants and antipsychotics for children |

Continued

**Table 1** Continued

| Study | Authors | Country | Time frame | Participants, setting | Study design | Interaction | Outcomes |
|---|---|---|---|---|---|---|---|
| 74 | Esmaily et al[60] | Iran | Not reported | 112 general physicians were randomised in two groups: (1) outcome-based educational intervention for rational prescribing and (2) concurrent CME programme in the field of rational prescribing | Randomised trial | Effect of outcome and retinal prescribing | Rational prescribing improved in some of the important outcome-based indicators. No difference between two arms of the study |
| 76 | Parikh et al[55] | USA | 2014 | Descriptive, cross-sectional analysis of Open Payments data and 9 638 825 payments to physicians and paediatricians from 1 January to 31 December 2014 | Cross-sectional | Comparison of PSR interactions between paediatricians and other specialists; among subspecialties of paediatrics. | Paediatricians get fewer gifts from PSR than internists. There is variation among subspecialties for extent of interaction. |
| 78 | Chressanthis et al[74] | USA | Not reported | Clinical decisions of 72 114 physicians were statistically analysed using prescription data | Survey | Effect of restricting PSRs on clinical practice and knowledge | Restricting PSRs affected information flow about drugs, both negative and positive. |

We excluded 2000 records as they were not relevant (n=1641), not original research (n=269), about medical students (n=4) and non-medical (eg, ecological and econometric; n=86). PSRs, pharmaceutical sales representatives.

### Pharmaceutical representative speakers

Sponsored lectures/symposia of pharmaceutical companies influenced behaviour of the attendees leading to the attendees prescribing more drugs from the sponsoring companies without sufficient evidence supporting superiority of those drugs.[56 57] The majority of attending physicians failed to identify inaccurate information about the company drug.[18]

### Honoraria and research funding

Physicians who received money to attend pharmaceutical symposia or to perform research requested formulary addition of the company's drug more often than other physicians. This association was independent of many confounding factors[61] (table 2). Brief encounters with PSRs and receipt of honoraria or research support were predictors of faculty requested change in hospital formulary.[68]

### Conference travel

Pharmaceutical company-sponsored conference travels to touristic locations have quantifiable impact on the prescribing rational of attendees. A significant increase (three times) in the prescribing rate of two company drugs was observed after the physicians attended a company-sponsored symposium with all their expenses covered. Despite this significant difference in the prescribing patterns, physicians insisted there was no impact on their prescribing behaviour.[57]

### Industry-paid lunches

Most physicians received invitations for dinners[9] and free lunches.[10 21 35 43] Clerks, interns and junior residents attended more company-sponsored lunches than senior residents.[10] Pharmaceutical companies also sponsored departmental lunches during journal clubs.[39] There was no significant association between attending industry-paid lunches[37] and dinners[9] and formulary request for that company's drug (table 2).

### CME sponsorship

Physicians who attended company-sponsored CME events had more positive attitudes towards and inclination to prescribe the branded drugs.[28 34 43 67 69–71] We found that physicians who refused CME sponsorship were seen to prescribe higher proportion of generics and lower expenditure medicines when compared with physicians who attended CMEs.[22]

### DISCUSSION

We report that there is widespread interaction between the pharmaceutical industry and physicians.[9–11 13 42 50] Interactions are in the form of personal communications, free gifts such as drug samples, sponsored meals, sponsored conference travel, funding for research and CMEs and honoraria.[9 21 22 31 36 42] The frequency of these interactions is comparable between residents and physicians.[10 21 22 36 42] However, the amount and type of gifts vary with the position

**Table 2** Impact of physician–pharmaceutical industry interaction on physician

| # | Attitudes | Prescribing behaviour | Knowledge | Formulary requests | Quality of evidence (GRADE) |
|---|---|---|---|---|---|
| Gifts | Receiving higher number of gifts associated with belief that PSR (pharmaceutical representative) have no impact on their prescribing behaviour[1 14 39] | – | – | – | Moderate |
| Drug samples | Positive attitude towards the drug industry and the representatives[11 21 34] | Higher prescription of the company drug[21 41] | – | – | High |
| Pharmaceutical representative speakers | – | Irrational prescribing[16 18 34] | Inability to identify false claims[16] | Increased prescription of sponsor's drug[24] | High |
| Honoraria and research funding | Positive attitude towards sponsor's drug[60] | – | – | Increased prescription of sponsor's drug[24] | Low |
| Conference travel | – | Significant increase in prescribing of sponsor drug[18] | – | Increased prescription of sponsor's drug[24] | Low |
| Industry-paid lunches | Positive attitude towards sponsor's drug[14 34] | Significant increase in prescribing of sponsor drug[62] | – | Increased formulary request for company drug[11 21] | High |
| CME sponsorship | Positive attitude towards sponsor's drug[24 65] | Avoidance of industry-sponsored CME associated with more rational prescribing habits[33] | | | Moderate |
| Interaction withPSR | Positive attitude towards PSR drugs[1 11 14 58] | Higher prescription of the company drug[24] | Positive correlation between the physicians' prescribing cost and the information provided by the drug representative during the interaction[26] | Increased prescription of sponsor's drug[24] | High |

However, there was a significant association between attending industry-paid lunches and increased prescription of branded drugs.[52 53 72]

of the physician in medical hierarchy, specialisation and location of practice.[10 13 31 38 42 43 50 58 62] In general, trainees (residents and interns) are treated with more drug samples, stationery items and free meals than senior physicians.[10 13] Senior physicians usually avail of sponsored conferences/ trips, research funding, honoraria and CME events. The extent of these interactions varies with academic versus non-academic institutions: non-academic hospitals record more interactions than others.[31 38 42 50 55] The majority of the physicians do not believe that they are affected by PSR interactions.[1 10 32 33 35 37 43 59] However, a sizeable percentage in various surveys responded in the affirmative when asked whether they thought that their peers are vulnerable.[1 20 21 37 43]

### Policies and educational intervention

The relationship of physicians with patients is of a fiduciary nature. Hence, activities that might affect that relationship by altering physicians' clinical behaviour are not acceptable. Physician–pharmaceutical industry and PSR interactions may put the trust of patients in physicians at risk. Interaction with pharmaceutical industry and PSRs begins early in the physicians' career. Trainees are exposed to pharmaceutical industry marketing and promotional techniques from the initial years of their medical education, which impact their prescribing behaviour in future. Overall, trainees, that is, residents and interns, are more vulnerable to pharmaceutical industry and PSR interactions than senior physicians[11 41 62] Physicians are susceptible to pharmaceutical industry and PSR interactions, which influences their clinical decision making leading to greater prescriptions of branded drugs over low-cost generic medicines and increasing healthcare costs.[22 47 52 53 72] Therefore, there is need to institute and implement stringent policies curtailing physician–pharmaceutical industry and PSR relationships, as well as educational programmes to increase awareness. Previous reports have indicated that implementing policies and conducting educational programmes are effective in increasing awareness of physician's attitudes towards pharmaceutical industry and PSR interactions.[54 59 60 73–83]

### Strengths and limitations of the study

A major strength of this study is that it is a large, up-to-date systematic review of studies exploring the effects of physician and pharmaceutical industry representative interactions and residents in different settings (eg, academic and primary care). Another strength of this study is the use of Cochrane and GRADE methodologies for conducting a review and assessing the quality of the studies. Moreover, we performed an extensive search in three databases and the grey literature. Some of the limitations of this review are related to the included studies, as some did not provide evidence for the significance of their findings or had varying study designs and outcomes, which made it impossible to conduct a meta-analysis. Also, the included studies were subject to risk of bias related to the lack of validity of outcome measurement and inadequate handling of significant potential confounders.

### Future implications

Pharmaceutical industry and PSR interactions compromise the objectivity of the physicians. Educating physicians and increasing regulation of pharmaceutical industry and PSR interactions may lower the likelihood of prescribing new non-superior industry drugs and irrational prescription behaviour. Further studies are required to evaluate the impact of pharmaceutical industry and PSR interactions on physicians over time and the benefits of various intervention-based education programmes on the clinical and ethical behaviour of the physicians.

**Contributors** All authors have contributed equally and have substantial contributions to the conception or design of the work: FF for the acquisition, analysis and interpretation of data for the work and drafting the work, EU and WF for revising it critically for important intellectual content and all authors contributed to final approval of the version to be published and agreed to be accountable for all aspects of the work in ensuring that questions related to the accuracy or integrity of any part of the work are appropriately investigated and resolved.

**Competing interests** None declared.

**Provenance and peer review** Not commissioned; externally peer reviewed.

**Data sharing statement** Any data relevant to a published article will be made available alongside the article when published.

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
