## [Reviewer comments · BMJ Open]

ARTICLE DETAILS

TITLE (PROVISIONAL)	Interactions between physicians and the pharmaceutical industry generally and sales representatives specifically and their association with physicians attitudes and prescribing habits: A systematic review
AUTHORS	Fickweiler, Freek; Fickweiler, Ward; Urbach, Ewout

VERSION 1 - REVIEW

REVIEWER	Joel Lexchin York University Canada
REVIEW RETURNED	15-Mar-2017

GENERAL COMMENTS	This paper is a systematic review that looks at interactions between physicians and the pharmaceutical industry and the consequences of those interactions. The authors have searched a number of databases from 1992 to 2016 to identify studies that meet their inclusion criteria. As it stands I have concerns about this paper that are mainly related to its methodology and whether it provides any new information. What the authors have shown is that interactions are frequent, that most doctors do not believe that they are influenced by the interactions, although they are not as confident on this point about their colleagues, and that the interactions often lead to increased prescribing of the company's products. While all of this is concerning none of this information is particularly new. At the end of the article (page 8, lines 33-35) the authors state that these interactions lead to irrational prescribing but, while this is likely, they actually have not made this case. Some of the English in this manuscript is stilted and it should be thoroughly edited by someone who has English as a first language. Page 2, line 15: The figure of \$89.5 billion comes from reference 5 but reference 5 is just citing other references. The authors should cite the primary source not a secondary source. Page 2, line 28: What "administration" are the authors referring to? Page 2, line 32: What "controversy" are the authors referring to? Page 2, line 51: The authors need to make it clear if they just included practising physicians or also included physicians in training. Did they just look at studies conducted in the community setting or did they also include studies done in the hospital setting? Page 3, Search strategy: Did the authors search for grey literature?
---

	If not why? Did the authors consult the systematic reviews on this topic done by Lexchin (1993), Wazana (2000) and Spurling (2010) as sources for studies to include? Page 3, line 18: Why did the authors choose 1992 as a starting point? If I read the authors correctly they are saying that they did not find studies before 1992 that investigated the physician-pharmaceutical company representative relationship. This is not the case - see the 2010 review by Spurling et al in PLoS Medicine. Page 3, lines 18-23: How were these search terms combined? Were they the same for each database? Page 3, lines 26-28: The review protocol should be summarized in the Methods. What data was extracted from each article? How were discrepancies resolved between reviewers? Page 4, line 16: Figure 1 says that 500 records were identified from other sources. What were those other sources? The authors need to briefly summarize the reasons why 2000 records were excluded, e.g., not original research, etc. Page 4, Characteristics of included studies: What was the geographical location of the studies and how many were done in the community and hospital settings? Page 4, lines 39-40: The authors said that participants were physicians but medical students are not physicians. Page 7, lines 46-50: The authors need to identify which countries have put legislation and policies into effect, what these consist of and whether they have been effective. Page 18, table: Why do the authors say "physicians" for some studies and "doctors" for others? This table should also say what country the study was conducted in and whether it was done in a hospital or community setting.
--	--

REVIEWER	Alan Maynard Department of Health Sciences University of York York YO10 5DD England
REVIEW RETURNED	18-Mar-2017

GENERAL COMMENTS	The literature about the relationship between physicians and the pharmaceutical industry is extensive and fragmented. This paper provides a systematic review of this material and, as usual, finds a modest number of studies of good quality. With pharmaceutical sales efforts there is always the problem of separating the beneficial educational effects of exposure against the problem that industry investments are primarily focused on sales and profit seeking. The paper summarises problems associated with the latter very well, and confirms the findings in related literature in books and journal material. When published this paper would provide excellent teaching material and the basis of improved evaluation of a complex and controversial issue
---

	In the authors' summary it would be useful to note some additional points It is to be noted that with nurse prescribing the problems exposed in the physician literature is being replicated. Also slow progress on increasing reporting and transparency international remains an issue
--	--

REVIEWER	Erwin Loh Monash University and Monash Health, Australia
REVIEW RETURNED	24-Mar-2017

GENERAL COMMENTS	Page 1 of 27 ----- Line 15 in PDF - sentence missing full stop 37 - spell put PSR 40 - spell out CME Page 2 ----- Line 3 - under strengths and limitations, you just described the study again. You should state why your description is a strength, not just describe it. Also the sentences are in dot point, not in paragraphs and full sentences. Line 13- "The relationship between physicians and the pharmaceutical industry has evoked heated debate since decades" - this sentence grammar is problematic - you may want to rewrite. Maybe change to "for many decades". Line 15 - add a "the" before pharmaceutical industry Line 23 - "evidence, while their colleagues more than themselves are susceptible to PSR marketing strategies" - this sentence is hard to understand. Do you mean "evidence. Physicians on the other hand believe their colleagues are more susceptible to PSR marketing strategies than themselves"
---

Line 28 - "However, while administration's proposals for deregulatory reforms of Big Pharma are increasing, scientific evidence rigourously examining this controversy are needed." This sentence has both grammar and spelling mistakes. Change to "administrative proposals", and I would avoid using the term "Big Pharma" unless you define and source it. It's spelt "rigorously".

Line 31 - "address" is "addresses"

Inclusion/ exclusion criteria

Line 40 - Did you factor in the significance ($p < 0.05$) of the studies as a criterion, and if that's not relevant explain why.

Also did you consider sample size of studies as a criterion?

Page 3

Line 17 - "in" should be "over"

Line 18 - "found" should be "find"

Line 23 - Any comment on reliability/ standardisation between the 2 independent reviewers who were involved in inclusion/exclusion?
Was there cross-checking?

Page 4

line 34 "number" -> "numbers"

line 37 - remove "report"

Page 6

Line 12 - "Sponsored lectures/symposia of pharmaceutical companies influenced behavior of the attendees, as they prescribed more drugs of the industry without sufficient evidence supporting the drug's superiority"

I think rephrase it to:

"Sponsored lectures/symposia of pharmaceutical companies influenced behavior of the attendees leading to the attendees prescribing more drugs from the sponsoring companies without sufficient evidence supporting superiority of those drugs" (if this is what you mean)

I'm not sure I understand the significance of the next sentence:

"The majority of attending physicians failed to identify inaccurate information about the company drug" - why is this an issue?

line 23 - this is an interesting issue - clinical trials - clearly if you do a clinical trial, you will need that drug on the formulary - I'm not sure clinical trials should be grouped in the same category. Industrial sponsorship of research staff, and other research support, may or may not be relevant, but pharmaceutically sponsored clinical trials is a huge area that may be a bit different to PSR influence.

line 28 - "Physicians considered company funded clinical trials with skepticism albeit their prescribing behavior was affected favoring the company's drug" - not sure what this means either. Do you mean physicians who prescribe drugs as part of the trial or outside the trial? Before, during or after? Are the physicians observing the trial? Needs some context.

line 52 - change "Pharmaceuticals" to "Pharmaceutical companies"

Page 7

line 5 - spell out CME when first time using the acronym - I know we all know what it is, but good to explain

Nice discussion section - good summary.

Line 43 - "More the amount and monetary value of the interactions, Considering that physicians have a social contract with society at large to provide unbiased and altruistic service, this is an alarming observation." - makes no sense, looks like an error - please fix.

Line 55 - add a "a" before "fiduciary" - also, you should provide a reference to a legal source for this.

Page 8

Line 15 - remove double full-stops ".." change to one.

Line 25 - other limitations - significance of the findings of the individual studies? You provide evidence for significance in some but not all

Also, is there a risk that only positive studies are published, and therefore found by your review. Did you find any negative studies, showing the opposite or no effect? If so, how many, and how significant? I suspect not many or none as they don't get published.

Line 35 - missing full-stop between "cost" and "Educating".

Page 18

Table 1 - would be good to know which ones had significant findings, the sample size of each one, and the country of origin here - at least I'm interested. But not essential.

Page 23

Table 2 - So here you have pooled the studies into themes, and then given a p-value, with a reference. This section is tricky - are you saying the other studies have no p-values, or that their p-values show insignificance? Is the number of studies significant? It's problematic. I'm not sure I like the presentation and what this is actually trying to say. Maybe just include studies that have significant findings here?

Page 25

Since you refer to PRISMA here, you should probably say that your methods conform to PRISMA in your methods in the paper. I know you refer to it in Figure one.

Final thoughts

This review would have been interesting if you reduced the number by being stricter, pooled similar studies with significant results, and do a meta-analysis on the pooled data, if that's possible. It's hard to know how significant your findings are besides being a commentary, as discussed above.

Table 2 is key - you need to explain how you have come up with the themes from the different studies, why they are there, which ones are significant, which are not, why you include them in each theme etc - hope this makes sense. As it is, some themes have a p value, some have OR/CI, some have nothing. Would be interesting to know the n - sample size.

	Overall, an important piece of work, and a great summary of the current evidence base on this important issue.
--	--

VERSION 1 – AUTHOR RESPONSE

Reviewer: 1

9. This paper is a systematic review that looks at interactions between physicians and the pharmaceutical industry and the consequences of those interactions. The authors have searched a number of databases from 1992 to 2016 to identify studies that meet their inclusion criteria. As it stands I have concerns about this paper that are mainly related to its methodology and whether it provides any new information. What the authors have shown is that interactions are frequent, that most doctors do not believe that they are influenced by the interactions, although they are not as confident on this point about their colleagues, and that the interactions often lead to increased prescribing of the company's products. While all of this is concerning none of this information is particularly new. At the end of the article (page 8, lines 33-35) the authors state that these interactions lead to irrational prescribing but, while this is likely, they actually have not made this case.

We appreciate the careful review and constructive suggestion you made, and it changed our statement that these interactions lead to irrational prescribing.

However, as you stated, we believe that we made it likely, and therefore changed it to "PSR interactions...are likely to result in irrational prescribing behavior and increasing healthcare costs (see lines 314-315). We used Cochrane and GRADE methodologies for conducting and assessing the quality of evidence of this systematic review. This systematic review provides new and up-to-date information on many categories of physicians to investigate and demonstrate the influence of PSR on doctors as widely as possible (see Methodology/Discussion, see lines 86-144 and 300-312).

10. Some of the English in this manuscript is stilted and it should be thoroughly edited by someone who has English as a first language.

We appreciate your comment and have thoroughly edited the manuscript to meet your comments.

11. Page 2, line 15: The figure of \$89.5 billion comes from reference 5 but reference 5 is just citing other references. The authors should cite the primary source not a secondary source.

We apologize for citing the secondary source. We have changed the reference to the primary source and have added another one with updated data to support our statement. The primary source is: "MedAdNews. Table insert on the top 50 pharma companies 2010. Data provided by CegeDIM Strategic Data, September 2010." The second reference that we have added with updated data that

supports our statement is: "Mack J. Pharma promotional spending in 2013. Pharma marketing news. May 2014, volume 13, number 5." (see references 7 & 8).

12. Page 2, line 28: What "administration" are the authors referring to?

We agree that these lines are troublesome to read. We were referring to healthcare administrators aiming to reduce the negative impact of physicians' interaction with pharmaceutical companies by implementing policies restricting free samples, meals, gifts, industry supplied promotional materials, and meetings with pharmaceutical company representatives. We have rewritten them, taking into account your critique and the comments of Reviewer 3 (see lines 81-85).

13. Page 2, line 32: What "controversy" are the authors referring to?

We agree that this sentence is hard to understand. The controversy we tried to refer to is that physicians believe their colleagues are more susceptible to PSR marketing strategies than themselves. We have rewritten the lines to clarify ourselves (see lines 78-85), and, also, to take into account the comments of Reviewer 3.

14. Page 2, line 51: The authors need to make it clear if they just included practising physicians or also included physicians in training. Did they just look at studies conducted in the community setting or did they also include studies done in the hospital setting?

We understand the need for clarification of the setting and participants. Both practising physicians and physicians in training were included: "Junior residents received ... material than residents." (see lines 163-164) as well as studies in community and hospital settings: "Participants working ... nonhospital practices" (see lines 169-173).

15. Page 3, Search strategy: Did the authors search for grey literature? If not why?

We did search for grey literature, and we included 500 articles identified in the grey literature (theses/dissertations) considered relevant to include in our review. We only found a modest number of studies of good quality here, an issue that is also pointed out by Reviewer 2. This is summarized in Figure 1 ('other sources'). We have added this information in the methods (see lines 128-130).

16. Did the authors consult the systematic reviews on this topic done by Lexchin (1993), Wazana (2000) and Spurling (2010) as sources for studies to include?

We did consult the abovementioned systematic reviews as sources for studies to include and have added the references to our revised manuscript (see references 2, 27-28).

17. Page 3, line 18: Why did the authors choose 1992 as a starting point? If I read the authors correctly they are saying that they did not find studies before 1992 that investigated the physician-pharmaceutical company representative relationship. This is not the case - see the 2010 review by Spurling et al in PLoS Medicine.

We chose this starting point, because other reviews already addressed the studies that investigated physician-pharmaceutical company representative relationships before 1992. As you already stated, Spurling et al reviewed these studies. We have changed this part to: "Databases were not searched before 1992, as these studies were already investigated in an earlier review." (see lines 124-125).

18. Page 3, lines 18-23: How were these search terms combined? Were they the same for each database?

A complete search strategy of one database is added as a supplementary information file. The search strategy was employed for searching PubMed and was adapted for other databases (see Supplementary information file – Search strategy).

19. Page 3, lines 26-28: The review protocol should be summarized in the Methods. What data was extracted from each article? How were discrepancies resolved between reviewers?

We would like to thank you for the critical analysis of our manuscript. Details of our review protocol has been added in the Methods section (see lines 86-144).

20. Page 4, line 16: Figure 1 says that 500 records were identified from other sources. What were those other sources? The authors need to briefly summarize the reasons why 2000 records were excluded, e.g., not original research, etc.

We did search for grey literature, and we included 500 articles identified in the grey literature (theses/dissertations) considered relevant to include in our review. We only found a modest number of studies of good quality, an issue that is also pointed out by Reviewer 2. This is summarized in Figure 1 ('other sources'). We have added this information in the methods, as mentioned at point 15. A brief summarization of the reason of exclusion has been added in the Results section (see lines 150-152).

21. Page 4, Characteristics of included studies: What was the geographical location of the studies and how many were done in the community and hospital settings?

We understand the need for clarification of the location and setting of the included studies. Therefore, Table 2 has been changed accordingly.

22. Page 4, lines 39-40: The authors said that participants were physicians but medical students are not physicians.

We agree. For clarity reasons, we removed these parts in the manuscript.

23. Page 7, lines 46-50: The authors need to identify which countries have put legislation and policies into effect, what these consist of and whether they have been effective.

This is a very interesting idea. We appreciate your comment on this matter. We referred to The Sunshine Act which was enacted in 2010 in the USA, as the first Congressional involvement to prevent undue influence of pharmaceutical companies on physicians and protect the public interest. Moreover, the WHO and the Health Action International have reported on interventions to counter promotional activities.

The evidence presented in that report suggested that interventions such as industry self-regulation and guidelines for sales representatives are not effective, while education about drug promotion might influence physician attitudes. At that time, in 2005, the report called for research on interventions that could affect doctors' behavior (reference: Norris P, Herxheimer A, Lexchin J, et al. Drug promotion—what we know, what we have yet to learn—reviews of materials in the WHO/HAI database on drug promotion. 2005. EDM Research Series No. 032.). The focus of our study was to review the physicians and PSR interactions, so we did not explore which countries put legislation and policies into effect, what these consist of and whether they have been effective. Further studies may explore this very interesting idea. We have rewritten this section (see lines 270-281).

24. Page 18, table: Why do the authors say "physicians" for some studies and "doctors" for others? This table should also say what country the study was conducted in and whether it was done in a hospital or community setting.

We will change "doctors" to physicians, as this might introduce unnecessary confusion for the reader. We understand the need for clarification of the location and setting of the included studies. Therefore, Table 2 has been changed accordingly.

Reviewer: 2

The literature about the relationship between physicians and the pharmaceutical industry is extensive and fragmented. This paper provides a systematic review of this material and , as usual, finds a modest number of studies of good quality. With pharmaceutical sales efforts there is always the problem of separating the beneficial educational effects of exposure against the problem that industry investments are primarily focused on sales and profit seeking. The paper summarises problems associated with the latter very well, and confirms the findings in related literature in books and journal material. When published this paper would provide excellent teaching material and the basis of improved evaluation of a complex and controversial issue. In the authors' summary it would be useful to note some additional points. It is to be noted that with nurse prescribing the problems exposed in the physician literature is being replicated. Also slow progress on increasing reporting and transparency international remains an issue.

We appreciate your careful review, constructive suggestions and positive comments. We agree with your additional points of slow progress on increasing reporting and transparency and the replication of the findings in non-physicians such as nurse prescribing. Although our study is focused on physicians, these are excellent suggestions. We have added this in our summary (see lines 259-263): "It is further noted ... such as nurses ... irrational prescribing in non-physicians."

Reviewer: 3

Page 1 of 27

25. Line 15 in PDF - sentence missing full stop

26. 37 - spell put PSR

27. 40 - spell out CME

We have changed the abovementioned comments.

Page 2

28. Line 3 - under strengths and limitations, you just described the study again. You should state why your description is a strength, not just describe it. Also the sentences are in dot point, not in paragraphs and full sentences.

We agree with the reviewer and have rewritten the Strengths and Limitations, taken into account your comments (see lines 300-312).

29. Line 13- "The relationship between physicians and the pharmaceutical industry has evoked heated debate since decades" - this sentence grammar is problematic - you may want to rewrite. Maybe change to "for many decades".

We have changed it into "The relationship ... for many decades" (see line 73).

30. Line 15 - add a "the" before pharmaceutical industry

This has been changed.

31. Line 23 - "evidence, while their colleagues more than themselves are susceptible to PSR marketing strategies" - this sentence is hard to understand. Do you mean "evidence. Physicians on the other hand believe their colleagues are more susceptible to PSR marketing strategies than themselves"

We meant the latter and have changed it accordingly (see lines 78-79).

32. Line 28 - "However, while administration's proposals for deregulatory reforms of Big Pharma are increasing, scientific evidence rigoursly examining this controversy are needed." This sentence has both grammar and spelling mistakes. Change to "administrative proposals", and I would avoid using the term "Big Pharma" unless you define and source it. It's spelt "rigorously".

This has been changed (see lines 81-85).

33. Line 31 - "address" is "addresses"

This has been changed.

Inclusion/ exclusion criteria

34. Line 40 - Did you factor in the significance ($p < 0.05$) of the studies as a criterion, and if that's not relevant explain why. Also did you consider sample size of studies as a criterion?

We did factor in the significance and sample sizes of the studies as criteria (see the Methodology). We included the significant studies from Table 1 in Table 2.

Page 3

35. Line 17 - "in" should be "over"

This sentence has been changed.

36. Line 18 - "found" should be "find"

This sentence has been changed.

37. Line 23 - Any comment on reliability/ standardisation between the 2 independent reviewers who were involved in inclusion/exclusion? Was there cross-checking?

We added standardisation and cross-checking in our Methods section (see lines 86-144).

Page 4

38. line 34 "number" -> "numbers"

This has been changed.

39. line 37 - remove "report"

This has been changed.

Page 6

40. Line 12 - "Sponsored lectures/symposia of pharmaceutical companies influenced behavior of the attendees, as they prescribed more drugs of the industry without sufficient evidence supporting the drug's superiority" I think rephrase it to: "Sponsored lectures/symposia of pharmaceutical companies influenced behavior of the attendees leading to the attendees prescribing more drugs from the sponsoring companies without sufficient evidence supporting superiority of those drugs" (if this is what you mean)

We would like to thank the reviewer for this suggestion and have changed it in our manuscript (see lines 208-209).

41. I'm not sure I understand the significance of the next sentence: "The majority of attending physicians failed to identify inaccurate information about the company drug" - why is this an issue?

We added this sentence to create awareness by the readers, that the pharmaceutical industry promotes (selective) information about their drug. This is an issue, because most attending doctors fail to identify (this as) inaccurate information/framing. This makes them more prone to prescribing in favor of the company drug by the interactions of pharmaceutical sales representatives. For clarity reasons, we have rewritten this part (see lines 210-213).

42. line 23 - this is an interesting issue - clinical trials - clearly if you do a clinical trial, you will need that drug on the formulary - I'm not sure clinical trials should be grouped in the same category. Industrial sponsorship of research staff, and other research support, may or may not be relevant, but pharmaceutically sponsored clinical trials is a huge area that may be a bit different to PSR influence.

This is an excellent comment. We meant that physicians who received money to attend pharmaceutical symposia or to perform research requested formulary addition of the company's drug more often than other physicians. The money that the physicians received was not to perform research for that company's drug. This association was independent of many confounding factors.

We rephrased this sentence: "Physicians who received money to attend pharmaceutical symposia or to perform research requested formulary addition of the company's drug more often than other physicians, This association was independent of many confounding factors." (see lines 216-218).

43. line 28 - "Physicians considered company funded clinical trials with skepticism albeit their prescribing behavior was affected favoring the company's drug" - not sure what this means either. Do you mean physicians who prescribe drugs as part of the trial or outside the trial? Before, during or after? Are the physicians observing the trial? Needs some context.

For clarity reasons, we have deleted this sentence.

44. line 52 - change "Pharmaceuticals" to "Pharmaceutical companies"

This has been changed.

Page 7

45. line 5 - spell out CME when first time using the acronym - I know we all know what it is, but good to explain

This has been changed.

46. Nice discussion section - good summary.

Thank you.

47. Line 43 - "More the amount and monetary value of the interactions, Considering that physicians have a social contract with society at large to provide unbiased and altruistic service, this is an alarming observation." - makes no sense, looks like an error - please fix.

This has been changed.

48. Line 55 - add a "a" before "fiduciary" - also, you should provide a reference to a legal source for this.

We changed this and added legal sources as a references (see line 307, reference 87 & 88).

Page 8

49. Line 15 - remove double full-stops ".." change to one.

This has been changed.

50. Line 25 - other limitations - significance of the findings of the individual studies? You provide evidence for significance in some but not all.

We added this as a limitation (see line 308).

51. Also, is there a risk that only positive studies are published, and therefore found by your review. Did you find any negative studies, showing the opposite or no effect? If so, how many, and how significant? I suspect not many or none as they don't get published.

We did not find negative studies in our search. This might, indeed, be caused by the publication bias that you mentioned.

52. Line 35 - missing full-stop between "cost" and "Educating".

This has been fixed.

Page 18

53. Table 1 - would be good to know which ones had significant findings, the sample size of each one, and the country of origin here - at least I'm interested. But not essential.

We have added sample size and country of origin in Table 1 and provided an overview of significant findings and quality of evidence in Table 2 .

Page 23

54. Table 2 - So here you have pooled the studies into themes, and then given a p-value, with a reference. This section is tricky - are you saying the other studies have no p-values, or that their p-values show insignificance? Is the number of studies significant? It's problematic. I'm not sure I like the presentation and what this is actually trying to say. Maybe just include studies that have significant findings here?

We agree, thank you for this very good suggestion. We have added significant findings and strength of evidence in Table 2 . We have changed it to clarify its meaning, to highlight the most significant findings and quality of evidence (see Table 2).

Page 25

55. Since you refer to PRISMA here, you should probably say that your methods conform to PRISMA in your methods in the paper. I know you refer to it in Figure one.

This has been changed (see lines 91-92).

Final thoughts

56. This review would have been interesting if you reduced the number by being stricter, pooled similar studies with significant results, and do a meta-analysis on the pooled data, if that's possible. It's hard to know how significant your findings are besides being a commentary, as discussed above. Table 2 is key - you need to explain how you have come up with the themes from the different studies, why they are there, which ones are significant, which are not, why you include them in each theme etc - hope this makes sense. As it is, some themes have a p value, some have OR/CI, some have nothing. Would be interesting to know the n - sample size.

We appreciate the careful review and comments of the reviewer. We believe that our revised manuscript has greatly improved because of your comments and the others, thank you.

57. Overall, an important piece of work, and a great summary of the current evidence base on this important issue.

Thank you. We hope that the revised manuscript makes it a more valuable and easier read.

VERSION 2 – REVIEW

REVIEWER	Joel Lexchin York University Canada
REVIEW RETURNED	In 2015-2016 Joel Lexchin received payment from two non-profit organizations for being a consultant on a project looking at indication based prescribing and a second looking at which drugs should be distributed free of charge by general practitioners. In 2015 he received payment from a for-profit organization for being on a panel that discussed expanding drug insurance in Canada. He is on the Foundation Board of Health Action International.

GENERAL COMMENTS	06-May-2017
-------------

REVIEWER	Erwin Loh Monash University, Australia
REVIEW RETURNED	15-May-2017

GENERAL COMMENTS	Thank you to the authors for addressing the comments raised by the editors. I have no further comments - the authors have addressed my concerns adequately.
---

VERSION 2 – AUTHOR RESPONSE

Reviewer 1

2. There are still problems with the English in this manuscript and it should be thoroughly edited by someone who has English as a first language.

We appreciate your comment and have thoroughly edited the manuscript by someone who has English as a first language (Allen C. Clermont, Boston, USA) to meet your comments.

3. Page 4, line 76: When the authors say "attitudes" do they mean the belief about whether or not interactions influence them, about whether or not the interactions are acceptable?

We mean the belief about whether or not PSR interactions influence physicians.

Revision: (page 4, line 78): However, the evidence determining whether or not PSR interactions influence physicians are divided and contradictory.

4. Page 4, line 81: What controversy is the authors referring to?

The controversy we tried to refer to is that physicians believe their colleagues are more susceptible to

PSR marketing strategies than themselves. We have rewritten these lines to clarify ourselves.

Revision: (page 4, line 81): The majority of the physicians do not believe that they are affected by PSR interactions 1, 13, 14, 15, 20 22, 25, 34.

4. Page 4, line 82: When the authors say "deregulatory reforms" do they mean reforms that would remove some governmental authority over the industry?

We mean reforms that would remove some governmental authority over the industry.

Revision (page 4, line 83) : However, while administrative proposals for deregulatory reforms that would remove some governmental authority over the industry are increasing,

5. Page 4, line 84: What is the question that the authors are investigating?

We appreciate your comment. We mean "controversy", not "question".

Revision: (page 4, line 86): This review addresses this controversy by critically and systemically evaluating the evidence on the impact of PSR interactions on physicians.

6. Page 4, line 85: Here the authors state that they are looking at the impact of interactions on the attitudes of physicians. In the Methods the authors state that the outcomes are attitudes, knowledge and behavior. Also in this sentence the authors only refer to PSRs but the Results list multiple different kinds of interactions aside from just interacting with PSRs.

We agree that this sentence may be unclear. The result section describes the different types of exposure between physicians and the PSR such as meeting with drug representatives, participating in pharmaceutical-sponsored CME event, receiving travel funding, free drug samples, industry-provided meals, gifts to the individual and active presentation of industry-related information to the physician. These multiple different kinds of exposure are all related to PSR interaction. We changed the sentence on page 4, line 85 (see revision).

Revision: This review addresses this controversy by critically and systemically evaluating the evidence on the impact of PSR interactions on physicians.

7. Page 4, line 90: What do the authors mean by "as per standardization in the protocol"?

Agree, this sentence is confusing.

Revision (line 90): Two independent reviewers assessed selected articles as per inclusion/exclusion criteria,

8. Page 4, line 91: When the authors say "shortlisted them for writing the review" do they mean inclusion in the review?

We tried to say that the reviewers shortlisted articles for the review and cross checked each other before including in the review.

Revision: Two independent reviewers assessed selected articles as per inclusion/exclusion criteria, shortlisted them for writing the review and cross-checked each other.

9. Page 4, line 98: What do the authors mean by "a comparator"?

We changed this sentence for clarity reasons.

Revision: (line 98) Types of studies: cross sectional studies, cohort studies, randomized trials and survey designs comparing an intervention of interest on at least one facet of extent, impact on behavior and attitude

10. Page 4, lines 99-100: On the next page the authors list exclusion criteria and this sentence should be removed here.

Thank you for this suggestion.

Revision: removed lines 99-100.

11. Page 5, lines 109-114: In listing the outcomes the authors are including three different types: attitudes, knowledge and behaviour. Are the authors looking at changes in each of these outcomes or just whether attitudes, knowledge and behaviour affect the frequency of interactions? This goes back to my earlier question - what "controversy" are the authors investigating?

We kindly refer to the answers for questions 4 and 6. We revised this sentences to be more clear.

12. Page 5, lines 113-114: What is a "lower level of interaction"?

We mean "no interaction".

Revision: Type of control: no interaction

13. Page 5, line 125: Reference 27 refers to the Spurling study that searched for articles up to February 2008 so I'm not clear why the authors did not start at that point rather than 1992.

We believe that the focus of the excellent review by Spurling et al. is different from the study that we used as a starting point of our review. The objective of the Spurling study was to examine the relationship between exposure to information from pharmaceutical companies and the quality, quantity, and cost of physicians' prescribing. The objective of our study and the study that we used as a starting point is to explore the frequency of physician and pharmaceutical industry interactions, their impact on physicians' attitude, knowledge and behavior

14. Page 5, line 128: The supplementary file only describes the search strategy for one database whereas the sentence implies that it gives the search strategies for all the databases. In addition, searching for grey literature is different than searching databases. How did the authors search for theses and dissertations?

We mention this in the method section: "The search strategy included Pubmed, Embase, Cochrane library and Google scholar electronic databases. Google scholar electronics databases were used for searching theses and dissertations. We agree that the sentence about the supplementary file is not clear.

Revision: The supplementary information file provides the full details for one database

15. Page 6, line 139: What information was extracted from the selected studies. How many authors were involved for each study and how were discrepancies resolved? Did the authors do any statistical analyses? What questions were they investigating? What statistical tests did they use and what software?

We have updated the information that was extracted from the selected studies to the main text of the manuscript (see revision). Two reviewers assessed in duplicate and independently the study design, types of interventions, outcomes assessed, and outcome measures in each eligible study. Disagreements were resolved by discussion or adjudication by a third reviewer (described in lines 141-142). The main question was to explore the frequency of physician and pharmaceutical industry interactions, their impact on physicians' attitude, knowledge and behavior. Eligibility criteria are described in lines 96-128.

Revision: The information extracted from the selected studies included type of study, study design, type of PSR interaction and type of outcome.

16. Page 6, line 140: Do the authors mean that they assessed the agreement with respect to what studies should be read in full?

We changed this sentence for clarity reasons.

Revision: Disagreements were resolved by discussion or adjudication by a third reviewer.

17. Page 6, line 146: The Methods section lists three types of outcomes of the interactions - attitudes, knowledge and behaviour but the Results are not organized along these lines. A good deal of the material in the Results talks describes the nature of the interactions and differences between different groups of doctors but does not actually deal with changes in attitudes, knowledge or behaviour.

This observation reflects the process of the study. When we initiated the study and designed the method section, we were interested in the outcomes of the interactions; attitudes, knowledge and behavior. However, when we conducted the study, it appeared that the majority of eligible studies deal with the nature of interactions and differences between different groups of doctors.

18. Page 6, line 158: In presenting the results the authors need to put them into context of when the research in the studies was conducted. For instance, reference 40 about what program directors of internal medicine residencies allow was published in 1992 but there is no indication in the text that this study is 25 years old.

This is a very interesting suggestion. However, the focus of our review was to describe the impact of PSR interactions on physicians. We did not search the databases before the publication date of 1992 because these studies were investigated by another review that searched databases for studies before publication date 1992. This review does not state when the research in the studies is conducted. We prefer to use publication date for the studies to prevent overlap with studies that have been already investigated by another review. Therefore, further research could focus on the impact of PSR interactions on physicians over time by investigating the exact dates of when the research in the studies is conducted.

Revision: Further studies are required to evaluate the impact of PSR interactions on physicians over time

19. Page 6, lines 161-163: What kind of differences were there in the frequency of interactions and gifts between the different groups?

Junior residents received twice as much free drug samples from PSR interactions than senior residents. PSR interactions were significantly higher at the beginning of residency (lines 177-179). Physicians in academic or hospital-based practice settings had less PSR interactions and significantly

lower prescribing costs than physicians in nonacademic and nonhospital practices (lines 186-188).

20. Page 6, line 168: What do the authors mean by "greater encounters"?

This is indeed confusing. We changed the sentence.

Revision: Attending physicians and physician specialists had more PSR interactions and received more numbers of medical samples and promotional material than residents.

21. Page 7, line 171: Do the authors mean doctors working solely in the public sector?

As far as the studies that we cite describe this, these are indeed doctors working solely in the public sector.

22. Page 8, lines 211-213: This sentence is poorly written and in addition it is not describing a result but rather offering an opinion and should be in the Discussion.

Agree. We removed lines 211-213.

23. Page 9, line 245: The outcomes of interest as stated by the authors in the Methods were attitudes, knowledge and behaviour and the Discussion should start with a summary of the results for each of these three outcomes.

We kindly disagree with this suggestion. The discussion starts with a summary of our main findings, followed by the description of the policies and educational intervention, strengths and limitations and future implications. We believe that we should start the discussion with a summary of the results of our main findings.

24. Page 9, lines 259-263: This is a review of physician interactions and the last two sentences in this paragraph should be deleted.

We agree with this comment and deleted the last sentences in this paragraph.

Revision: removed lines 259-263.

25. Page 9, line 268: If the authors are going to talk about the impact of interactions on healthcare costs they need to provide a reference for this statement.

Agree. We changed this sentence.

Revision: Considering that physicians have a social contract with society at large to provide unbiased and altruistic service, this is an alarming observation.

26. Page 9, line 268 to page 10, line 274: This is a very long sentence and very hard to follow. Also the Sunshine Act was not put in place to curb activities that abuse the role of physicians but to report on payments to physicians.

Agree. We removed lines 268-274.

27. Page 10, lines 274-275: See the previous comment. The Sunshine Act is not meant to prevent undue influence or protect the public interest. It is purely a piece of legislation that requires reporting. How the reporting is analyzed is a different question and not part of the legislation.

Agree. We removed lines 274-275.

28. Page 10, lines 286-287: Studies about medical students were not part of this review and therefore if the authors want to talk about interactions between medical students and the industry they need to cite pertinent references.

Agree. We changed this sentence.

Revision: Interaction with PSRs begins early in the physicians' career.

29. Page 10, lines 292-294: Adding drugs of companies that physicians have interacted with to formularies may be either good or bad depending on the therapeutic utility of the drug, its cost and whether it is more or less valuable than something already on the formulary. The authors are assuming that adding drugs to formularies is inherently negative.

That's true.

We removed this sentence.

30. Page 10, line 303: As I said earlier, much of the literature that the authors have cited does not deal with the impact of the interactions on any of these three measures.

Agree. We changed this sentence.

Revision: A major strength of this study is that is a large up-to-date systematic review of studies exploring the effects of physician and pharmaceutical industry representative interactions and residents in different settings (e.g. academic, primary care).

31. Page 11, lines 311-312: Why is the time frame a limitation of the study?

Agree, we deleted this sentence in the discussion.

32. Page 11, line 314: The authors focus here on interactions with PSRs but they have cited many studies that deal with other types of interactions.

We agree. We changed this sentence.

Revision: PSR interactions compromise the objectivity of the physicians.

33. Page 44, table: Besides giving the date of publication the table should also give the dates during which the research was conducted.

This is a very interesting suggestion. However, the focus of our review was to describe the impact of PSR interactions on physicians. We did not search the databases before the publication date of 1992 because these studies were investigated by another review that searched databases for studies before publication date 1992. This review does not state when the research in the studies is conducted. We prefer to use publication date for the studies to prevent overlap with studies that have been already investigated by another review. Therefore, further research could focus on the impact of PSR interactions on physicians over time by investigating the exact dates of when the research in the studies is conducted.

Revision: Further studies are required to evaluate the impact of PSR interactions on physicians over time

VERSION 3 - REVIEW

REVIEWER	Joel Lexchin York University Canada In 2015-2016 Joel Lexchin received payment from two non-profit organizations for being a consultant on a project looking at indication based prescribing and a second looking at which drugs should be distributed free of charge by general practitioners. In 2015 he received payment from a for-profit organization for being on a panel that discussed expanding drug insurance in Canada. He is on the Foundation Board of Health Action International.
REVIEW RETURNED	29-Jun-2017

GENERAL COMMENTS	1. The authors have not made it clear whether they are investigating the effects of all forms of interactions between doctors and industry or just the effects of interactions with PSRs. For instance, in the Abstract the Objective says “explore the frequency and of physician and pharmaceutical industry interactions” but the Data Synthesis only refers to PSRs. Similarly, in the text, the Introduction talks mainly about PSRs but the Methods describes a wide variety of types of interactions. (See my comment later on about this.) The Discussion opens by summarizing all of the various types of interactions but the “policies and educational intervention” and “future implications” sections only deal with interactions with PSRs.2. I think that the authors have misinterpreted one of earlier comments. For the studies that they have included in their review they should not only document the date of publication but they should also document the time period when the research was carried out as there can be a considerable gap between conducting the research and publishing it. Table 1 should have an additional column labeled something like “Timeframe of research” and for the Steinman article the entry would be “Spring 1999”.3. There are still a number of minor grammatical errors that need to be corrected. Here are examples from the Abstract and Introduction:a. Page 2, line 10: “Its” should be “their”b. Page 2, line 31: “influences” should be “influence”c. Page 2, line 31: delete “the” before “physicians”d. Page 2, line 40: delete “the” at the end of the linee. Page 2, line 42: “prescription” should be “prescribing”f. Page 4, line 14, “are” should be “is”4. Page 4, line 27: It is still not clear what “this controversy” refers to.5. Page 4, line 42: This should be rewritten as “cross-checked their decisions about inclusion/exclusion with each other”.6. Page 5, lines 3-10: The authors have listed many different types of exposure but since the purpose of this study seems to be to investigate interactions between doctors and PSRs they need to expressly make the link between doctors interacting with PSRs and the receipt of travel funding, industry-provided meals, etc. Many of these types of gifts do not necessarily require interacting with PSRs.7. Page 16, lines 41-43: The part of the sentence reading “and significantly lower...nonhospital practices” does not fit with the title of
--

	this subsection. 8. Page 17, lines 13-15: The sentence “We found that...and their gifts” does not fit with the title of this subsection. 9. Page 17, line 26: The subsections starting with Gifts describe the effect of these various types of interactions on prescribing and the titles of these subsections should reflect the material in them. (Except the “Gifts” subsection doesn’t describe the influence of gifts. If there wasn’t any evidence of the effect of gifts then the authors should briefly state that.)
--	--

VERSION 3 – AUTHOR RESPONSE

1. The title is awkward. I would suggest a slightly longer but more accurate title: Interactions between physicians and the pharmaceutical industry generally and sales representatives specifically and their association with physicians attitudes and prescribing habits: A systematic review

Revision: changed title: Interactions between physicians and the pharmaceutical industry generally and sales representatives specifically and their association with physicians attitudes and prescribing habits: A systematic review

2. Page 2, lines 8-11: I believe that the authors are looking at more than just the frequency of interactions. Rewrite as follows: The objective of this review is to explore interactions between physicians and the pharmaceutical industry including sales representatives and their impact on...

Revision: changed lines 8-11: The objective of this review is to explore interactions between physicians and the pharmaceutical industry including sales representatives and their impact on

3. Page 2, lines 10-11: Attitude, knowledge and behavior about what - value of the interactions, appropriateness of prescribing, value of drugs, etc.?

Revision: changed lines 10-11: physicians’ attitudes and prescribing habits.

4. Page 18, lines 33-37: The sentence starting "Conference registration fees..." should be moved to the section on gifts.

Revision: moved lines 33-37 sentence starting “Conference registration fees..” to the section on gifts.

5. Page 18, line 45: The sentence starting "Most of the physicians..." should be moved to the section on gifts.

Revision: moved line 45 sentence starting “Most of the physicians..” to the section of gifts.

VERSION 4 – REVIEW

REVIEWER	Joel Lexchin York University Canada
REVIEW RETURNED	11-Jul-2017

GENERAL COMMENTS	I have a few more minor suggestions for this manuscript. The title is awkward. I would suggest a slightly longer but more accurate title: Interactions between physicians and the pharmaceutical industry generally and sales representatives specifically and their association with physicians attitudes and prescribing habits: A systematic review Page 2, lines 8-11: I believe that the authors are looking at more than just the frequency of interactions. Rewrite as follows: The objective of this review is to explore interactions between physicians and the pharmaceutical industry including sales representatives and their impact on... Page 2, lines 10-11: Attitude, knowledge and behavior about what - value of the interactions, appropriateness of prescribing, value of drugs, etc.? Page 18, lines 33-37: The sentence starting "Conference registration fees..." should be moved to the section on gifts. Page 18, line 45: The sentence starting "Most of the physicians..." should be moved to the section on gifts.
--

VERSION 4 – AUTHOR RESPONSE

We are very grateful for the suggestions of reviewer 1.

1. The title is awkward. I would suggest a slightly longer but more accurate title: Interactions between physicians and the pharmaceutical industry generally and sales representatives specifically and their association with physicians attitudes and prescribing habits: A systematic review

Revision: changed title: Interactions between physicians and the pharmaceutical industry generally and sales representatives specifically and their association with physicians attitudes and prescribing habits: A systematic review

2. Page 2, lines 8-11: I believe that the authors are looking at more than just the frequency of interactions. Rewrite as follows: The objective of this review is to explore interactions between physicians and the pharmaceutical industry including sales representatives and their impact on...

Revision: changed lines 8-11: The objective of this review is to explore interactions between physicians and the pharmaceutical industry including sales representatives and their impact on

3. Page 2, lines 10-11: Attitude, knowledge and behavior about what - value of the interactions,

appropriateness of prescribing, value of drugs, etc.?

Revision: changed lines 10-11: physicians' attitudes and prescribing habits.

4. Page 18, lines 33-37: The sentence starting "Conference registration fees..." should be moved to the section on gifts.

Revision: moved lines 33-37 sentence starting "Conference registration fees.." to the section on gifts.

5. Page 18, line 45: The sentence starting "Most of the physicians..." should be moved to the section on gifts.

Revision: moved line 45 sentence starting "Most of the physicians.." to the section of gifts.